# An L-Shaped Three-Level and Single Common Element Sparse Sensor Array for 2-D DOA Estimation

**DOI:** 10.3390/s23146625

**Published:** 2023-07-23

**Authors:** Bo Du, Weijia Cui, Bin Ba, Haiyun Xu, Wubin Gao

**Affiliations:** 1Institute of Information Engineering, PLA Strategic Support Force Information Engineering University, Zhengzhou 450001, China; zzu202122332015625@gs.zzu.edu.cn (B.D.); cuilink_work@sina.com (W.C.); xuhaiyun1995@163.com (H.X.); 2School of Cyber Science and Engineering, Zhengzhou University, Zhengzhou 450002, China; 3Research Office, Information Center, Beijing 100000, China; zhushilei3620@163.com

**Keywords:** sensor array, wireless communication, direction of arrival (DOA) estimation, array signal processing, L-shaped three-level, single common element sparse array (LTSESA)

## Abstract

The degree of freedom (DOF) is an important performance metric for evaluating the design of a sparse array structure. Designing novel sparse arrays with higher degrees of freedom, while ensuring that the array structure can be mathematically represented, is a crucial research direction in the field of direction of arrival (DOA) estimation. In this paper, we propose a novel L-shaped sparse sensor array by adjusting the physical placement of the sensors in the sparse array. The proposed L-shaped sparse array consists of two sets of three-level and single-element sparse arrays (TSESAs), which estimate the azimuth and elevation angles, respectively, through one-dimensional (1-D) spatial spectrum search. Each TSESA is composed of a uniform linear subarray and two sparse subarrays, with one single common element in the two sparse subarrays. Compared to existing L-shaped sparse arrays, the proposed array achieves higher degrees of freedom, up to $4Q_{1}Q_{2}+8Q_{1}-5$, when estimating DOA using the received signal covariance. To facilitate the correct matching of azimuth and elevation angles, the cross-covariance between the two TSESA arrays is utilized for estimation. By comparing and analyzing performance parameters with commonly used L-shaped and other sparse arrays, it is found that the proposed L-shaped TSESA has higher degrees of freedom and array aperture, leading to improved two-dimensional (2-D) DOA estimation results. Finally, simulation experiments validate the excellent performance of the L-shaped TSESA in 2-D DOA estimation.

## 1. Introduction

As we all know, direction-of-arrival (DOA) estimation is an important part of array signal processing. One-dimensional (1-D) DOA estimation can only estimate the azimuth angles but cannot estimate the elevation angles. Fortunately, two-dimensional (2-D) DOA estimation can solve the above problem. Now, the 2-D direction of arrival estimation problem is widely used in radar, internet of vehicle (IOV) and the fifth-generation (5 G) mobile communications [1,2,3,4,5,6]. And many algorithms have been developed to solve the problem of DOA estimation, such as improved reduced dimension MUSIC (IRD-MUSIC), the reduced-dimension multiple signal classification algorithm, and so on [7,8,9,10,11,12]. Compared to a 2-D planar array, L-shaped sparse arrays have lower costs and better adaptability. They can be flexibly arranged and adjusted according to specific requirements to meet the demands of particular tasks. Therefore, developing a structurally more reasonable and higher degree of freedom L-shaped sparse array is a valuable proposition.

Moreover, the special geometry of the array used for DOA estimation also plays a crucial role in the performance of DOA estimation. The concept of sparse array design refers to the development of sparse arrays with higher degrees of freedom to reduce the hardware cost of DOA estimation and improve its performance. Specifically, the traditional L-shaped array consists of two parts. The subarrays in the *X*-axis and *Z*-axis are uniform linear arrays. The geometric structure of the uniform linear array (ULA) limits the DOA estimation performance of the traditional L-shaped array, which has low utilization of array elements, low uniform degrees of freedom (uDOF), small array aperture and poor DOA estimation ability. In order to increase the uniform degrees of freedom and expand the array aperture, many sparse arrays have been developed to solve the 1-D DOA estimation problem, such as minimum redundancy array (MRA) [13], nested array (NA) [14] and coprime array (CA) [15].

Subsequently, the sparse arrays are naturally applied to the 2-D DOA. The coprime planar array (CPA) [16] is represented by two parts, which contain the square of $T_{1}$ array elements or the square of $T_{2}$ array elements, respectively. Here, $T_{1}$ and $T_{2}$ are a set of coprime integers. CPA can estimate up to $min\{T_{1}^{2},T_{2}^{2}\}-1$ signals from the $T_{1}^{2}+T_{2}^{2}$ array elements. However, the application of 2-D planar sparse arrays is not flexible enough and has a relatively low degree of freedom. Compared with 2-D planar arrays, L-shaped sparse arrays generally have higher utilization of array elements [17,18,19]. Under a fixed number of array elements, more consecutive lags can be formed to improve the uDOF of the array and improve the performance. Therefore, this paper focuses on L-shaped sparse arrays. In the literature [20], the L-shaped coprime array (LCA) is used for 2-D DOA estimation. LCA consists of two parts, and each part contains $2M_{1}+M_{2}-1$ array elements. LCA can estimate the DOA of no more than $M_{1}M_{2}$ signals, and its array’s uDOF has much room for improvement. At this point, $M_{1}$ and $M_{2}$ are also a pair of coprime integers. The design of LCA is relatively simple, so the increase in degrees of freedom is not significant. The introduction of L-shaped optimized interleaved array’s (LOIAC) [21] configuration significantly improves the uDOF and array aperture of L-shaped sparse arrays. LOIAC consists of two parts, each containing $N_{1}+N_{2}$ array elements, where $N_{1}$ and $N_{2}$ are also a pair of coprime integers. Each part contains two subarrays, which can be estimated as $2\left(N_{1}+1\right)\left(N_{2}-\left\lfloor N_{1}\right\rfloor /2\right)+1$ signals. The uDOF of sparse arrays is often closely related to the number of subarrays. The single part of LOIAC only has two subarrays [22], so the array elements were not fully utilized. And the utilization of array elements, uDOF and array aperture of L-shaped sparse arrays still have room for improvement.

In order to enhance the geometric structure of the L-shaped sparse array and improve the performance of the array for 2-D DOA estimation, a new L-shaped sparse array named the L-shaped three-level and single common element sparse array (LTSESA) is proposed. Compared with other L-shaped sparse arrays, TSESA has more subarrays, higher utilization of array elements, and can form more consecutive lags, thereby increasing the uDOF of the array, expanding the aperture of the array, and improving the performance of 2-D DOA estimation. In general, the main research content of this paper is as follows:(1)This paper proposes a new L-shaped sparse array, named three-level and single common element sparse array. Its array element configuration has a complete mathematical expression.(2)According to the mathematical expression of TSESA, its uniform degrees of freedom expression can be derived.(3)We evaluate the performance of the popular L-shaped sparse array and the proposed L-shaped TSESA for 2-D DOA estimation, which fully demonstrates the superiority of the TSESA geometry.

The rest of this paper is structured as follows. The suggested L-shaped sparse array configuration is further explained in Section 2. The signal model based on an L-shaped TSESA is presented in Section 3. Section 3 also presents the technique for automatically matching elevation with anticipated azimuth angles. In Section 4, we present the simulation’s findings. Finally, Section 5 brings this paper to a conclusion.

We use bold lower- and upper-case letters to distinguish between vectors and matrices throughout the work. The transpose, complex conjugation and conjugate transpose are shown by the superscripts ${\left[•\right]}^{T}$, ${\left[•\right]}^{*}$ and ${\left[•\right]}^{H}$, respectively. The mathematical expectation is shown by $E\left[•\right]$. The operator for vectorization is $\mathrm{vec}\left(•\right)$. $\mathrm{diag}\left[•\right]$ denotes a matrix that is diagonal. The N×N identity matrix is denoted by $I_{N}$. Rounding up or down is represented by $\left\lfloor •\right\rfloor$ and $\left\lceil •\right\rceil$. ${\left|\left|•\right|\right|}_{F}^{2}$ denotes the Frobenius norm. ${\left[•\right]}^{†}$ denotes the Moore–Penrose pseudo-inverse operation.

## 2. The Configuration of L-Shaped Three-Level and Single Common Element Sparse Array

Because the existing L-shaped sparse array still has a large room for improvement, such as low utilization of array elements, less freedom, and smaller array aperture, in order to improve the performance of the L-shaped sparse array for 2-D DOA estimation, this section proposes an L-shaped three-level and single common element sparse array.

### 2.1. The Mathematical Expression of Array Element’s Position

The L-shaped array consists of two vertical parts, each of which is a three-level and single common element sparse array. The center of the intersection of two identical TSESA s is defined as the 0 element. Assume that the inter-element spacing is represented by *d*, where *d* is equal to a half wavelength. The array element arrangement structure of the TSESA is shown in Figure 1, which is composed of three subarrays and has a total of *M* array elements. The subarray 1 is a uniform linear array (ULA) with $Q_{1}$ elements. Both subarray 2 and subarray 3 are sparse linear arrays, which contain $Q_{1}$ and $Q_{2}+1$ elements, respectively. It is worth noting that subarray 2 and subarray 3 are connected by a single common array element at element $Q_{1}Q_{2}+4Q_{1}-3$, which is one of the characteristics of array TSESA. The following is the expression of the array element arrangement of the TSESA, where K, $K_{1}$, $K_{2}$ and $K_{3}$ represent the array, subarray 1, subarrays 2 and subarrays 3, respectively:(1)$$\left\{\begin{array}{c} K=K_{1}\cup K_{2}\cup K_{3}\\ K_{1}=\left\{md\left|m\in \left[0,Q_{1}-1\right]\right.\right\}\\ K_{2}=\left\{(2m+Q_{1}Q_{2}+2Q_{1}-1)d\left|m\in \left[0,Q_{1}-1\right]\right.\right\}\\ K_{3}=\left\{(mQ_{1}+Q_{1}Q_{2}+4Q_{1}-3)d\left|m\in \left[0,Q_{2}\right]\right.\right\} \end{array}\right.$$
where
(2)$$\left\{\begin{array}{c} Q_{1}=2\lfloor M/6\rfloor -1\\ Q_{2}=M-2Q_{1} \end{array}\right.$$

### 2.2. The uDOF and Array Aperture of TSESA

Uniform degrees of freedom and array aperture are important performance indicators for measuring sparse arrays for DOA estimation. The greater the uDOF, the more signals the array can estimate. The larger the array aperture, the higher the spatial resolution of the array. In general, we assume that the maximum number of consecutive lags of a sparse array is equal to the value of its uDOF.

Suppose that the positions of subarrays *R*, *O* and *P* are represented by sets $\left\{r_{1},r_{2},\ldots r_{R}\right\}$, $\left\{o_{1},o_{2},\ldots o_{O}\right\}$ and $\left\{p_{1},p_{2},\ldots p_{P}\right\}$, respectively. Then, the definition of the self-difference set $T_{self}$ and cross-difference set $T_{cross}$ can be expressed by the following two formulas.

**Definition** **1.**
*(Self-difference coarray): The self-difference coarray $T_{self}$ of the array supplied by array elements’ position set is defined as*

(3)
$$\begin{array}{cc} T_{self} & =\left\{r_{i}-r_{j}|i,j\in \left[1,R\right]\right\}\cup \left\{o_{i}-o_{j}|i,j\in \left[1,O\right]\right\}\\ \; & \cup \{p_{i}-p_{j}|i,j\in \left[1,P\right]\}. \end{array}$$



**Definition** **2.**
*(Cross-difference coarray): The cross-difference coarray $T_{cross}$ of the array supplied by array elements’ position set is defined as*

(4)
$$\begin{array}{cc} T_{cross} & =\{r_{i}-o_{j}|i\in \left[1,R\right],j\in \left[1,O\right]\}\\ \; & \cup \{r_{i}-p_{j}|i\in \left[1,R\right],j\in \left[1,P\right]\}\\ \; & \cup \{o_{i}-p_{j}|i\in \left[1,O\right],j\in \left[1,P\right]\}\\ \; & \cup \{o_{i}-r_{j}|i\in \left[1,O\right],j\in \left[1,R\right]\}\\ \; & \cup \{p_{i}-r_{j}|i\in \left[1,P\right],j\in \left[1,R\right]\}\\ \; & \cup \{p_{i}-o_{j}|i\in \left[1,P\right],j\in \left[1,O\right]\}. \end{array}$$


*Set T can be seen as the union of set $T_{self}$ and set $T_{cross}$. All lags and consecutive lags of the set T represent DOF and uDOF, respectively. The array aperture of the array is equal to max{T}−min{T}.*


As a result, the following lemma can be derived:

**Lemma** **1.**
*The range of consecutive lags of TSESA is that*

(5)
$$\begin{array}{c} T_{consecutive}=\left[3-2Q_{1}Q_{2}-4Q_{1},2Q_{1}Q_{2}+4Q_{1}-3\right]. \end{array}$$



**Proof.** See Appendix A.□

Compared with the existing L-shaped sparse array, it can be known that the L-shaped array proposed in this paper has higher degrees of freedom and array aperture as shown in Table 1 and Table 2.

## 3. The Signal Model and Estimation Method Used

### 3.1. The Signal Model

As shown in Figure 2, the L-shaped TSESA consists of two perpendicular TSESAs, located on the *X*-axis and *Z*-axis, respectively. The intersection center of two TSESAs is defined as 0. It is assumed that *K* signals hit the L-shaped TSESA from *K* directions ${\left(\theta_{k},\beta_{k}\right)}_{k=1}^{K}$. In this paper, the signal number is regarded as a priori information and can be estimated by the minimum description length (MDL) criterion [23]. These signals have the following characteristics:(1)The sources of these signals are located in the far field range of the array;(2)These signals are narrow-band signals;(3)These signals are irrelevant.

θ and β are used to represent the azimuth and elevation angles. As shown in Figure 2, the incident signal is emitted from the signal source towards the L-shaped array. The angle between the incident path of the signal and the *X*-axis of the array is defined as the azimuth angle. The angle between the incident path of the signal and the *Z*-axis of the array is defined as the elevation angle. Considering the characteristics of sparse configuration, the mutual coupling interference between sensor elements is very small, and the focus of this article is on the design of sparse arrays. Therefore, the model is constructed under ideal conditions, with negligible mutual coupling interference.

Based on the above communication environment, the signal reception model of the L-shaped TSESA located in the *X*-axis and *Z*-axis can be modeled as follows:(6)$$x\left(t\right)=\sum_{k=1}^{K}a_{X}\left(\theta_{k}\right)s_{k}\left(t\right)+n_{X}\left(t\right)=A_{X}s\left(t\right)+n_{X}\left(t\right),$$
(7)$$z\left(t\right)=\sum_{k=1}^{K}a_{Z}\left(\beta_{k}\right)s_{k}\left(t\right)+n_{Z}\left(t\right)=A_{Z}s\left(t\right)+n_{Z}\left(t\right).$$
where $s_{k}\left(t\right)$ denotes the *k*th incident signal; s(t) is a vector of all incident signals; n(t) represents the noise vector, where the noise is Gaussian white noise and is not related to the signals; and $a_{X}\left(\theta_{k}\right)$ and $a_{Z}\left(\beta_{k}\right)$ denote the azimuth or elevation steering vector of the *k*th incident signal, respectively. Specifically, the *w*th element in the steering vector can be expanded as follows:(8)$${\left[a_{X}\left(\theta_{k}\right)\right]}_{w}=e^{\frac{-j2\pi m_{w}d\sin\left(\theta_{k}\right)}{\lambda }},$$
(9)$${\left[a_{Z}\left(\beta_{k}\right)\right]}_{w}=e^{\frac{-j2\pi m_{w}d\sin\left(\beta_{k}\right)}{\lambda }}.$$
$m_{w}d$ represents the mathematical expression of the position of the *w*th element, and $m_{w}d$ belongs to K.

### 3.2. The Estimation of Azimuth Angles

Since the noise vector n(t) and the signal vector s(t) are not correlated, the autocorrelation operation is performed on the signal vector:(10)$$\begin{array}{c} R_{X}=E\left[x\left(t\right)x^{H}\left(t\right)\right]=A_{X}R_{s}{A_{X}}^{H}+{\sigma_{n}}^{2}I, \end{array}$$
(11)$$\begin{array}{c} R_{Z}=E\left[z\left(t\right)z^{H}\left(t\right)\right]=A_{Z}R_{s}{A_{Z}}^{H}+{\sigma_{n}}^{2}I. \end{array}$$
where $R_{s}=diag\left({\sigma_{1}}^{2},{\sigma_{2}}^{2},\ldots {\sigma_{K}}^{2}\right)$ and ${\sigma_{n}}^{2}$ represent the power of the incident signal and noise, respectively.

Then, by vectorizing $R_{X}$ and $R_{Z}$ respectively, we can obtain
(12)$$\begin{array}{c} x=vec\left(R_{X}\right)=P_{X}R+{\sigma_{n}}^{2}\bar{I}, \end{array}$$
(13)$$\begin{array}{c} z=vec\left(R_{Z}\right)=P_{Z}R+{\sigma_{n}}^{2}\bar{I}. \end{array}$$
where $R={\left[{\sigma_{1}}^{2},{\sigma_{2}}^{2}\ldots {\sigma_{K}}^{2}\right]}^{T}$ and $\bar{I}=\left[e_{1}^{T},e_{2}^{T},\ldots e_{M}^{T}\right]$. $e_{w}^{T}$ is a column vector, except the *w*-th element is equal to 1; the other elements are 0.

$P_{X}$ and $P_{Z}$ can be expanded as follows, and the ⊙ is the Khatri–Rao product:(14)$$\begin{array}{c} P_{X}={A_{X}}^{*}⊙A_{X}, \end{array}$$
(15)$$\begin{array}{c} P_{Z}={A_{Z}}^{*}⊙A_{Z}. \end{array}$$

The steering matrices $P_{X}$ and $P_{Z}$ represent T on the *X*-axis and *Z*-axis, respectively. T is generated by the self difference or mutual difference between subarrays 1, 2 and 3. x and z can be regarded as the incident signals received on T. But T is discontinuous, and we need to use continuous T. Therefore, we define *U* as the largest consecutive element in T, and use the subscript U to represent the largest continuous part in T.

In order to avoid the interference of spurious peaks on DOA estimation, the spatial smoothing technique [24,25] is introduced to deal with $x_{U}$:(16)$$\begin{array}{c} R_{temp}=x_{U}x_{U}^{H}. \end{array}$$

Then, $R_{ss}$ can be easily obtained by averaging all submatrices:(17)$$\begin{array}{c} R_{ss}=\frac{1}{L}\sum_{L=1}^{L}R_{temp}\left(m\right), \end{array}$$
(18)$$\begin{array}{c} R_{temp}=R_{temp}\left(m:m+l+1,m:m+l+1\right). \end{array}$$
where $L=\frac{l+1}{2}$ and 1≤m≤L.

Furthermore, through a series of MUSIC algorithms, the spatial spectrum function can be expanded as follows, where ${\bar{a}}_{XU}$ denotes the virtualized steering vector on the *X*-axis and $E_{N}$ denotes the noise subspace. By searching the spatial spectrum function, the azimuth angle θ can be successfully estimated:(19)$$\begin{array}{c} {\hat{\theta }}_{k}=\arg\underset{\theta }{min}\;{\bar{a}}_{XU}^{H}\left(\theta_{k}\right)E_{N}E_{N}^{H}{\bar{a}}_{XU}\left(\theta_{k}\right). \end{array}$$

### 3.3. The Estimation of Matched Elevation Angles

In the following sections, the azimuth information $\hat{\theta }=\left({\hat{\theta }}_{1},{\hat{\theta }}_{2},\ldots ,{\hat{\theta }}_{K}\right)$ estimated in the previous section is used to estimate the elevation angle β. By calculating the cross covariance of x(t) and z(t), it is effective to match the estimated azimuth and elevation angles.

By substituting the estimated azimuth information, the steering matrix of the array can be displayed in the following form:(20)$$\hat{A_{X}}=\left[a\left({\hat{\theta }}_{1}\right),a\left({\hat{\theta }}_{2}\right),\cdot \cdot \cdot ,a\left({\hat{\theta }}_{K}\right)\right].$$

The cross covariance of x(t) and z(t) is shown by
(21)$$\begin{array}{c} R_{XZ}=E\left[x\left(t\right)z^{H}\left(t\right)\right]=A_{X}R_{s}{A_{Z}}^{H}. \end{array}$$

Since each column element of $A_{X}$ corresponds to $A_{Z}$ one by one, the steering matrix $A_{Z}$ that needs to be estimated can be equivalent to its corresponding matrix $A_{X}$. By solving the following least squares problem, the steering matrix $A_{Z}$ can be successfully estimated [26]:(22)$$\begin{array}{c} {\hat{A}}_{Z}=\arg\underset{{\hat{A}}_{Z}}{min}{\left|\left|R_{XZ}-{\hat{A}}_{X}R_{s}A_{Z}^{H}\right|\right|}_{F}^{2}. \end{array}$$

The $R_{s}$ in the above formula can be obtained by eigenvalue decomposition of the covariance matrix $R_{X}$:(23)$$\begin{array}{c} R_{X}=E_{s}\Lambda_{s}E_{s}^{H}+E_{n}\Lambda_{n}E_{n}^{H}. \end{array}$$
where $\Lambda_{s}\in C^{K\times K}$ and $\Lambda_{n}\in C^{\left(M-1-K)\times (M-1-K\right)}$. Particularly, $\Lambda_{s}$ is a diagonal matrix of eigenvalues of $E_{s}$; $\Lambda_{n}$ is a diagonal matrix of eigenvalues of $E_{n}$; $E_{s}$ is the signal subspace eigenvector matrices; and $E_{n}$ is the noise subspace eigenvector matrices. By combining () and (Equation 23), $R_{s}$ can be estimated [26]:(24)$$\begin{array}{c} {\hat{R}}_{s}={{\hat{A}}_{X}}^{†}E_{s}\Lambda_{s}E_{s}^{H}{\left({{\hat{A}}_{X}}^{H}\right)}^{†}. \end{array}$$

Therefore, ${\hat{A}}_{Z}$ can be derived as the following mathematical expression:(25)$$\begin{array}{cc} {\hat{A}}_{Z} & ={\left({{\hat{R}}_{s}}^{-1}{\left({\hat{A}}_{X}\right)}^{†}R_{XZ}\right)}^{H}\\ \; & ={\left({\left({{\hat{A}}_{X}}^{†}E_{s}\Lambda_{s}E_{s}^{H}{\left({\hat{A}}_{X}^{H}\right)}^{†}\right)}^{-1}{{\hat{A}}_{X}}^{†}R_{XZ}\right)}^{H}. \end{array}$$
where ${\hat{A}}_{Z}\in C^{M\times K}$, and *M* cannot be less than *K* because of rank deficiency.

Therefore, the covariance matrix of the elevation angles can be calculated by the estimated ${\hat{A}}_{Z}$:(26)$$\begin{array}{c} {\hat{R}}_{Z_{k}}={\left[{\hat{A}}_{Z_{k}}\right]}_{:,k}{\left[{\hat{A}}_{Z_{k}}\right]}_{:,k}^{H}. \end{array}$$

By searching the spatial spectrum function as follows, the azimuth angle β can be successfully estimated. At this time, the estimated azimuth angles $\hat{\theta }=\left({\hat{\theta }}_{1},{\hat{\theta }}_{2},\ldots ,{\hat{\theta }}_{K}\right)$ and elevation angles $\hat{\beta }=\left({\hat{\beta }}_{1},{\hat{\beta }}_{2},\ldots ,{\hat{\beta }}_{K}\right)$ are matched one by one:(27)$$\begin{array}{c} {\hat{\beta }}_{k}=\arg\underset{\beta }{min}\;{\bar{a}}_{ZU}^{H}\left(\theta_{k}\right)E_{N}E_{N}^{H}{\bar{a}}_{ZU}\left(\theta_{k}\right). \end{array}$$

## 4. Numerical Experiments

In this section, we evaluate the differences in the 2-D DOA estimation performance of various array configurations through simulation experiments. The array configurations involved in the evaluation include the L-shaped ULA, L-shaped CA, L-shaped OIAC and L-shaped TSESA proposed in this paper. The array elements’ locations are shown in Table 3. In order to ensure the fairness of the comparison by controlling the variables, we set the number of array elements in the array to 23, and each signal has the same transmit power. The text uses the root mean square error (RMSE) curve to show the estimation performance of the arrays, and its mathematical expression is as follows:(28)$$\begin{array}{c} \mathrm{RMSE}=\sqrt{\frac{1}{2N_{s}K}\sum_{k=1}^{K}\sum_{N_{s}=1}^{N_{s}}{\left({\hat{\theta }}_{N_{s},k}-\theta_{k}\right)}^{2}+{\left({\hat{\beta }}_{N_{s},k}-\beta_{k}\right)}^{2}}. \end{array}$$
where $N_{s}$, ${\hat{\theta }}_{N_{s},k}$ and ${\hat{\beta }}_{N_{s},k}$ express the Monte Carlo times, azimuth and elevation estimation for the $N_{s}$-th trial of the *k*-th signal.

The definition of signal-to-noise ratio (SNR) is given by the following formula:(29)$$\begin{array}{c} \mathrm{SNR}=10\log_{10}\frac{\delta_{p}^{2}}{\delta_{n}^{2}}. \end{array}$$
where $\delta_{k}$ represents the power of the *k*th signal, and $\delta_{n}$ represents the noise power. In this paper, it is assumed that all sources are of equal power and all elements have similar noise.

### 4.1. Experiment 1

The main purpose of this experiment is to verify the effectiveness of the L-shaped TSESA proposed in this paper for two-dimensional DOA estimation, and to verify the ability to pair the estimated azimuth and elevation angles ${\left(\theta_{k},\beta_{k}\right)}_{k=1}^{K}$ one by one. The array structure of the L-shaped TSESA used is shown in Table 3. The number of snapshots is set to 200, and the SNR is set to 5 dB. The estimated three pairs of test signals are incident on the array from directions ${\left(-30{}^{\circ},50{}^{\circ}\right)}_{k=1}$, ${\left(20{}^{\circ},60{}^{\circ}\right)}_{k=2}$ and ${\left(40{}^{\circ},70{}^{\circ}\right)}_{k=3}$, respectively, and the test signal input is arranged in order. As shown in Figure 3, the spatial spectrum of the azimuth and elevation angles of the test signals is successfully drawn. The spectral peaks represent the estimated values, and the curves are clear, and the spectral peaks are clear too. The spatial spectra of the elevation angles are drawn one by one according to the order of the corresponding azimuth angles, such as Figure 3b–d. So the estimated azimuth and elevation angles are successfully paired one by one.

### 4.2. Experiment 2

The purpose of this experiment is to study the ability of each L-shaped array for the multi-signal estimation or underdetermined estimation. Since the array elements’ locations in the *X*-axis and *Z*-axis are the same, only the DOA estimation of the test azimuth angles is sufficient for evaluation. When the array is used for multi-signal estimation, the number of snapshots required is relatively high. Therefore, in this experiment, the number of snapshots is expanded to 1000, and the SNR is reduced to 0 dB. To maintain fairness in the comparison and control variables, we establish a configuration of 23 elements for all the arrays.

Since the uniform array cannot achieve underdetermined estimation, we use 11 directions as test signals, which are derived from 11 equal divisions from −60° to 60° as shown in the Figure 4a. The reason for using 11 signal sources in this case is because for an L-shaped uniform linear array (ULA), both its *X*-axis and *Z*-axis form uniform linear arrays, making it impossible to achieve underdetermined estimation. Taking the *X*-axis as an example, with 12 elements, we can estimate a maximum of 11 signal sources. L-shaped CA, L-shaped OIAC and L-shaped TSESA can achieve underdetermined estimation. The L-shaped CA can estimate up to $M_{1}M_{2}$ signals [20]. Since the $M_{1}$ and $M_{2}$ values of the L-shaped CA used for testing are 4 and 5, respectively, we use 20 directions as test signals as shown in Figure 4b. Similarly, the number of signal sources is limited to a maximum estimable value of 20 [20]. These directions are derived from 20 equal divisions from −60° to 60°. For L-shaped OIAC and L-shaped TSESA, we use 24 directions as test signals, which are derived from 24 parts from −60° to 60° as shown in Figure 4c,d. From this, it can be seen that for both L-shaped OIAC and L-shaped TSESA, the L-shaped array can achieve underdetermined estimation. The spatial spectra of four arrays’ azimuth angles for multi-signal estimation or underdetermined estimation are shown in Figure 4. Among them, the number of signals that can be estimated by L-shaped TSESA is not only more than that of L-shaped ULA and L-shaped CA, but also compared with L-shaped OIAC, the spatial spectrum of L-shaped TSESA is clearer, the estimation accuracy is higher, and the estimation effect is better.

### 4.3. Experiment 3

In order to verify the superior performance of the L-shaped TSESA for two-dimensional DOA estimation, this paper introduces RMSE as an effective indicator of performance verification. The smaller the RMSE, the more accurate the DOA estimation of the array and the better the performance. In Experiment 3, the number of snapshots is fixed at 200. SNR takes −10 dB as the starting point, 2 dB as the step, and 10 dB as the termination point. The signals used for estimation are $\left(-65{}^{\circ},70{}^{\circ}\right)$, $\left(15{}^{\circ},-40{}^{\circ}\right)$ and $\left(45{}^{\circ},-25{}^{\circ}\right)$, respectively. The number of array elements of the L-shaped ULA, L-shaped CA, L-shaped OIAC and L-shaped TSESA is 23, that is, the hardware cost. In this experiment, the value of the Monte Carlo times takes 500.

As shown in Figure 5, with the change of SNR, the signal environment becomes better, and the four RMSE curves show a downward trend, which indicates that the effect of the four L-shaped arrays for DOA estimation is gradually getting better. It is worth noting that, in the whole process, the RMSE curve representing the L-shaped TSESA is always lower than other curves because of more consecutive lags and larger array aperture [27]. This fully demonstrates that when the signal environment of the array is consistent with the same algorithm used, the L-shaped TSESA has the higher DOA estimation accuracy and the better performance. It also shows that the L-shaped TSESA array has a better effect under the same hardware cost.

### 4.4. Experiment 4

SNR is set at 5 dB in Experiment 4. The number of snapshots starts at 100, the step is 100, and the finish point is 1000. The estimation signals are $\left(-65{}^{\circ},70{}^{\circ}\right)$, $\left(15{}^{\circ},-40{}^{\circ}\right)$ and $\left(45{}^{\circ},-25{}^{\circ}\right)$. L-shaped ULA, L-shaped CA, L-shaped OIAC and L-shaped TSESA all have 23 array elements. As demonstrated in Figure 6, as the number of snapshots increases, the signal environment improves, and the four RMSE curves exhibit a lower trend, indicating that the array’s effect on DOA estimation is steadily improving. In this experiment, the value of the Monte Carlo times takes 500.

Through comparison, it can be clearly seen that the RMSE curve representing the L-shaped TSESA is always lower than the other curves. This signifies that the L-shaped TSESA has the higher DOA estimate accuracy and the better performance because of more consecutive lags and larger array aperture [27] when the signal environment of the array is consistent with the same algorithm utilized. It also demonstrates that the L-shaped TSESA has the best benefit when the number of array elements is kept constant.

## 5. Conclusions

In this paper, a new type of L-shaped sparse array is proposed for two-dimensional DOA estimation, called LTSESA. The characteristics of the array are that each part contains three subarrays; subarray 2 and subarray 3 have a single common array element. So the unique array structure design is more efficient and reasonable. The array has large uniform degrees of freedom, up to $4Q_{1}Q_{2}+8Q_{1}-5$, and an array aperture, which can improve the estimation accuracy and the underdetermined estimation ability of two-dimensional DOA estimation. In addition, the array has a complete mathematical expression. This advantage enables the rapid calculation of the sensor positions for LTSESA arrays of any size. Simulation results show that the array has better DOA estimation ability. In the future research work, we can try to further enhance the array structure and improve the performance indicators of L-shaped sparse arrays.

## Figures and Tables

**Figure 1 sensors-23-06625-f001:**
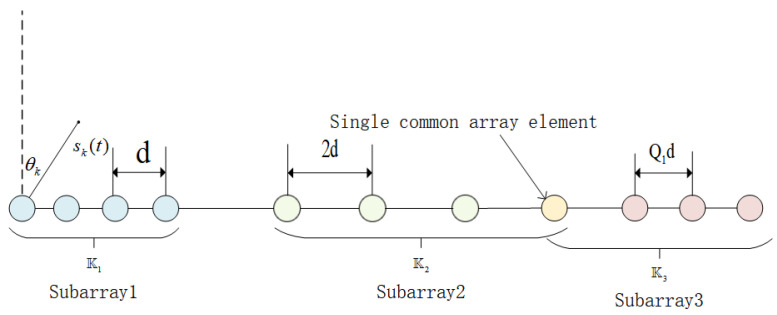
The geometry of TSESA.

**Figure 2 sensors-23-06625-f002:**
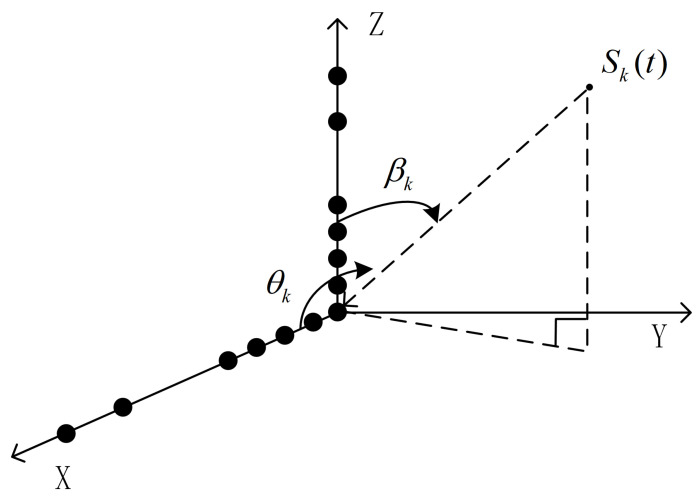
The geometry of L-shaped TSESA.

**Figure 3 sensors-23-06625-f003:**
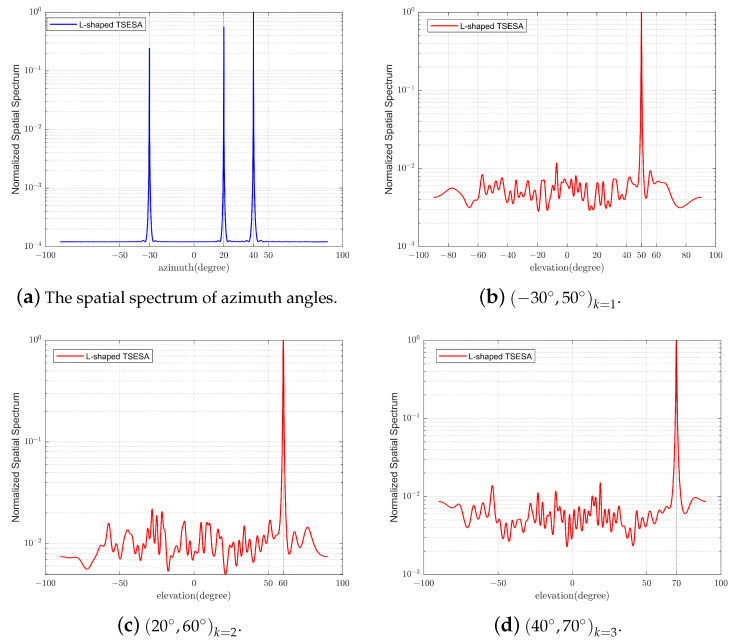
The spatial spectrum of azimuth and elevation angles.

**Figure 4 sensors-23-06625-f004:**
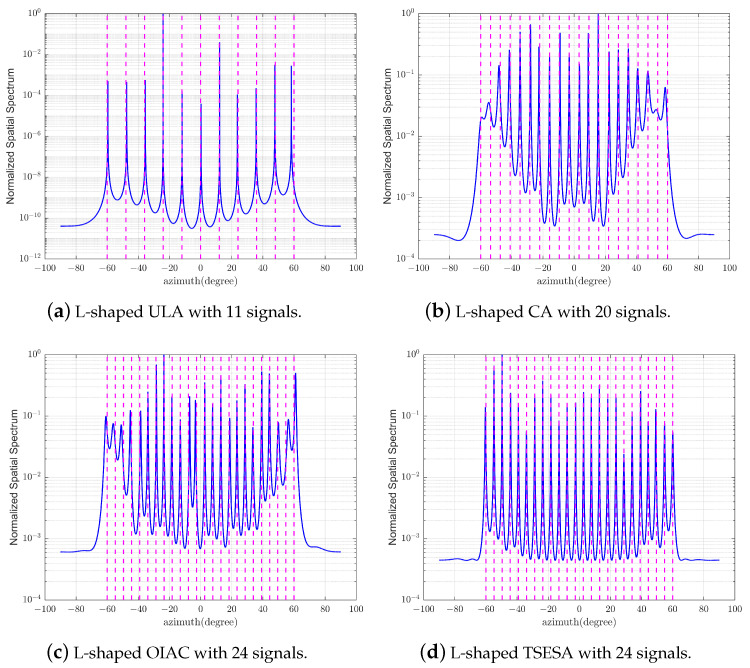
The spatial spectrum of azimuth angles for multi-signal estimation or underdetermined estimation.

**Figure 5 sensors-23-06625-f005:**
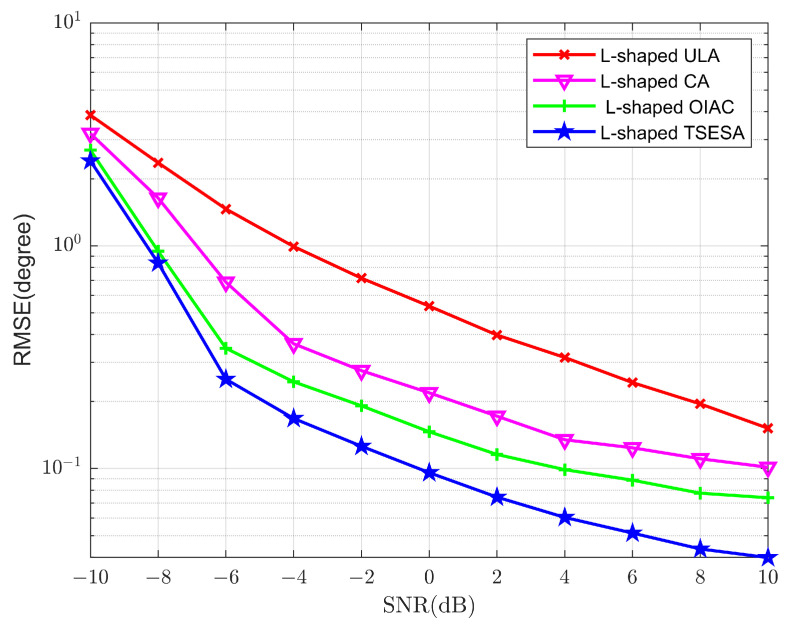
RMSE curves of DOA estimation versus SNR.

**Figure 6 sensors-23-06625-f006:**
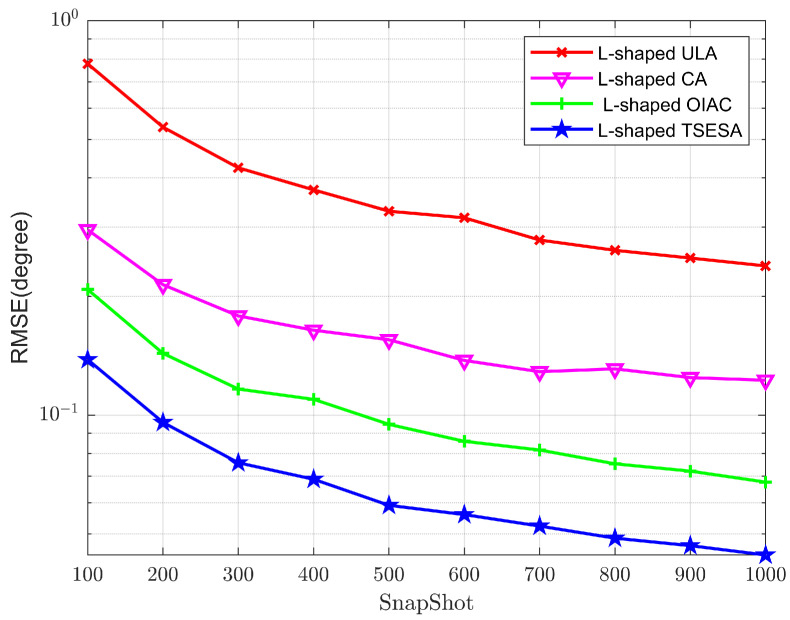
RMSE curves of DOA estimation versus snapshots.

**Table 1 sensors-23-06625-t001:** Consecutive lag numbers for different arrays.

Number of Array Elements	The Largest Number of Consecutive Lags (*X*-axis or *Z*-axis)			
	L-Shaped ULA	L-Shaped CA	L-Shaped OIAC	L-Shaped TSESA
23	23	47	65	91
29	29	65	97	127
35	35	95	133	195
41	41	125	177	255
47	47	167	225	331

**Table 2 sensors-23-06625-t002:** Aperture of the array for different arrays.

Number of Array Elements	The Maximum Aperture of the Array (*X*-axis or *Z*-axis)			
	L-Shaped ULA	L-Shaped CA	L-Shaped OIAC	L-Shaped TSESA
23	12	35	39	45
29	15	54	47	63
35	18	77	62	97
41	21	104	87	127
47	24	143	159	165

**Table 3 sensors-23-06625-t003:** The array elements’ locations for different arrays.

Different Arrays	Element Positions	
	Intersection Element’s Location	Other Elements’ Locations (*X*-axis or *Z*-axis)
L-shaped ULA	0	1, 2, 3, 4, 5, 6, 7, 8, 9, 10, 11
L-shaped CA	0	4, 5, 8, 10, 12, 15, 16, 20, 25, 30, 35
L-shaped OIAC	0	4, 8, 12, 16, 20, 14, 18, 19, 32, 34, 39
L-shaped TSESA	0	1, 2, 23, 25, 27, 30, 33, 36, 39, 42, 45

## Data Availability

The data that support the findings of this study are available upon request from the authors.

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
