# Peer review of "An L-Shaped Three-Level and Single Common Element Sparse Sensor Array for 2-D DOA Estimation"

_sensors, 2023, doi:10.3390/s23146625_

Round 1

Reviewer 1 Report

1. The novelty of the work should be clearly described in the abstract and conclusion using data and numbers to back up their claims. The abstract should be rewritten in the order: Problem, Objective, Method, Results, and Conclusion.

2. The related work section is not well written; the authors should insert the results of the previous related works and make a critical analysis by introducing the weaknesses or shortcomings of these works. 

3. I am attching the plagiarism report as it is showing more than 20% of similarity.

The manuscript requires moderate english proofreading as it has several grammatical errors. 

Reviewer 2 Report

1. the concept of the work considered by authors has not been justified in the text

2. DOA concept in sensor network should have been assisted by some mathematical modeling on ray line concept

3. Novel word should be removed from title as it does not have novelty in the current area

4. results are not convincing

5. need to justify the work proposed with modeling and issues of communication

Not sure

Reviewer 3 Report

The authors present a novel method to estimate the direction of arrival of signals in space. The proposed methodology includes a novel sensor arrangement and the related algorithm for DOA estimation.

The paper is well-written and has merit. The introduction and the state of the art are sufficiently clear to make the contributions evident.

I believe that the main room for improvement in this paper lies in experimental results. Simulations are fine to compare the novel TSESA against state-of-the-art methods, but an experimental assessment on a real case would enrich the article.

Finally, did the authors consider the introduction of optimality criteria to derive general methods for the sensor arrangement? Both in the introduction and in the conclusions the authors express the will to "optimize" the sensor array, but the paper lacks an optimization problem statement.

As minor remarks: 1) Please define properly Q1 and Q2 (line 44). 2) In Figure 2, shouldn't theta_k be defined in the X-Y plane?

Reviewer 4 Report

In this paper, the authors proposed a new L-shaped array and verified its effectiveness in DOA estimation. Here are some major concerns on the writing and the simulations before it can be accepted.

(1)  The motivation of adopting the L-shaped array need to be further polished. The logic of the first paragraph in the Introduction need to be improved.

(2)  As the authors claimed, the proposed L-shaped array realized higher DOFs than the existing L-shaped arrays. However, only one simulation tried to verify it. More importantly, in the experiment 2, the authors considered to use different sources to evaluate the estimated spectrum. However, it is obviously unfair. The authors did not demonstrate that the proposed array can estimate more DOAs than other arrays. Besides, the authors did not mention whether all the array adopt the same number of sensors.

(3)   It is well known that the L-shaped array suffers from the azimuth-elevation pairing problem. However, the authors did not mention how to solve this problem. Please introduce how to realize the pairing in detail.

(4)  The authors only compared with the performance among different arrays. Please consider the MSE lower bound, e.g.,

     [R1] Z. Zhang, Z. Shi and Y. Gu, "Ziv-Zakai Bound for DOAs Estimation," in IEEE Transactions on Signal Processing, vol. 71, pp. 136-149, 2023, doi: 10.1109/TSP.2022.3229946.

The motivation of adopting the L-shaped array need to be further polished. The logic of the first paragraph in the Introduction need to be improved.

Round 2

Reviewer 1 Report

The authors have incorporated all the suggestions in the revised manuscript. But it still requires minor editing of the English language.

But it still requires minor editing of the English language.

Reviewer 2 Report

1. observations are clarified to some extent

2. Result analysis could be improved